# Current Expertise, Opinions, and Attitude toward TNF-⍺ Antagonist Biosimilars among Physicians: A Self-Administered Online Survey in Western Switzerland

**DOI:** 10.3390/healthcare10112152

**Published:** 2022-10-28

**Authors:** Marko Krstic, Jean-Christophe Devaud, Farshid Sadeghipour, Joachim Marti

**Affiliations:** 1Institute of Pharmaceutical Sciences of Western Switzerland, University of Geneva, University of Lausanne, 1206 Geneva, Switzerland; 2Service of Pharmacy, Lausanne University Hospital and University of Lausanne, 1011 Lausanne, Switzerland; 3Center for Research and Innovation in Clinical Pharmaceutical Sciences, Lausanne University Hospital and University of Lausanne, 1011 Lausanne, Switzerland; 4School of Pharmaceutical Sciences, Department of Hospital Pharmacy, University of Geneva, 1206 Geneva, Switzerland; 5Faculty of Biology and Medicine, University of Lausanne, 1005 Lausanne, Switzerland; 6Center for Primary Care and Public Health (Unisanté), University of Lausanne, DESS, Health Economics Unit, 1010 Lausanne, Switzerland

**Keywords:** biosimilar adoption and commercialization, tumor necrosing factor-alpha/tnf-alpha/survey, cost savings, physician incentive plans

## Abstract

Tumor necrosis factor-alpha (TNF-⍺) antagonists are biological drugs with multiple authorized biosimilars. Biosimilars are becoming critical to the financial sustainability of health systems. Recent studies emphasize that physicians’ knowledge regarding biosimilars has not yet progressed sufficiently to overcome their concerns regarding biosimilars’ safety and efficacy. To assess the current knowledge, opinions, and attitudes toward TNF-⍺ antagonist biosimilars among postgraduate physicians and specialists, an anonymous, self-administered survey was implemented on SurveyMonkey between February and May 2022. The survey was validated through think-aloud interviews with senior and postgraduate physicians in rheumatology, gastroenterology, and immunoallergology, and a senior epidemiologist. Participant recruitment was conducted with the help of the physicians’ professional societies and departmental head physicians of two university hospitals in Western Switzerland. Most physicians felt more comfortable initiating a TNF-⍺ antagonist biosimilar in biologic-naive patients (BNPs) than switching patients stabilized on the original biologic (originator). However, most participants agreed that BNPs should start treatment with the biosimilar rather than the originator when available. Postgraduate physicians and specialists in rheumatology, gastroenterology, and immunoallergology who participated in this survey were familiar with TNF-⍺ antagonist biosimilars and were confident in prescribing them. Yet, they still preferred to avoid switching a patient already on the originator.

## 1. Introduction

The market for biologics continues to grow faster than the market for non-biologic medicines, and is expected to account for 30% of global pharmaceutical sales by 2025 [1,2,3]. The constant and growing craze for biologics has been putting pressure on health expenditure. In most countries, the arrival of biosimilars made it possible to considerably increase the number of patients benefiting from these innovative therapies while sparing the paying parties [3,4,5,6].

Biosimilars are biologics that are highly similar to a reference biologic (i.e., originator), with whom they share no clinically significant differences. As biologics, biosimilars are manufactured by biotechnology using living organisms such as bacteria, plants, or animal cells [7,8]. Biosimilars can only be registered on the market after the originator’s patent has expired, at a price that is 10 to 30% lower than that of the originator [3,9].

In this study, we will focus on the biosimilars of three tumor necrosis factor-alpha (TNF-⍺) antagonists licensed in Switzerland: infliximab, etanercept, and adalimumab. These biologics are used to treat inflammatory autoimmune diseases and possess multiple authorized biosimilars [3,10,11,12,13,14,15,16]. Their first biosimilars were approved by the EU in September 2013, January 2016, and March 2017, respectively [3]. As of December 2021, Switzerland was still among the European countries with the lowest adoption of TNF-⍺ antagonist biosimilars, with only a 27% market share in treatment days, compared with the European average of 60% [3].

A lack of knowledge and misunderstanding surrounding biosimilars have been highlighted in the early 2010′s as barriers to biosimilars’ acceptance by physicians, across multiple specialties [17,18,19,20]. Two systematic reviews that included studies from 2014 to 2019 reported that physicians’ expertise have been progressing but that objectively-measured knowledge was generally more limited than their self-assessed knowledge [21,22]. Essentially, the early skepticism from healthcare professionals about prescribing and/or administrating biosimilars has been decreasing thanks to their increased experience and ongoing education efforts [23]. However, the most recent studies emphasize that physicians' knowledge regarding biosimilars has not yet progressed enough for them to overcome their concerns regarding biosimilars’ safety, efficacy, and extrapolation of indications [23,24,25,26,27,28,29,30,31,32,33]. Altogether, these elements contribute to limiting the potential savings mentioned above and putting the financial sustainability of healthcare systems at risk [25].

To date, the exact reasons explaining the low uptake of TNF-⍺ antagonists biosimilars in Switzerland are still unclear [34]. The knowledge of Swiss prescribers regarding biosimilars has only been assessed in a 2019 survey, which reported that only 31% out of 93 Swiss specialists were very familiar with biosimilar medicines [27]. Hence, to better understand the potential barriers to the use of biosimilars in Switzerland, the present study aims to assess the current expertise, opinions, and attitude toward TNF-⍺ antagonist biosimilars among postgraduate physicians and specialists in Western Switzerland.

## 2. Materials and Methods

The questionnaire was sent to postgraduate physicians and specialists in rheumatology, gastroenterology, and immunoallergology, practicing in Western Switzerland, likely to prescribe TNF-⍺ antagonists and their biosimilars. Physicians from other specialties where TNF-⍺ antagonists are used only in response to autoimmune reactions from concomitant therapies such as oncology or hematology were not included. The survey was self-administered on SurveyMonkey (Momentive Inc., San Mateo, CA, USA), an online survey site [35].

The head physicians of the rheumatology, gastroenterology, and immunoallergology departments at two university hospitals were contacted to recruit hospital-based physicians. In parallel, the Swiss societies of Gastroenterology (SSG), of Rheumatology (SSR) and immunoallergology (SSAI) were contacted to recruit the specialists practicing in their offices or in other medical structures. The project was accepted by all parties, with the exception of the SSAI, one of the head physicians of the immunoallergology department and one of the head physicians of the gastroenterology department. Participation in the survey was voluntary and was only possible through links sent to the parties who agreed to help us recruit participants. No incentives were offered to increase the response rate.

The questionnaire was developed especially for this study by three hospital pharmacists and one health economist, based on previously published surveys from 2014 to 2020 [17,18,27,36,37]. The questions were developed in French, as this is the official language used in Western Switzerland. The first version of the survey was reviewed and refined by a senior epidemiologist, with expertise in survey design. The updated survey was then implemented on SurveyMonkey and technical functionalities were reviewed on the desktop and mobile versions by two authors.

Eight interviews using a think-aloud protocol were conducted by one of the pharmacists responsible for developing the questionnaire, which led to further changes to the questionnaire. Questionnaires were administrated on a desktop computer or a tablet to a senior physician from each specialty and five postgraduate physicians (one in rheumatology, one in immunoallergology, two in anesthesiology and one in internal medicine) to obtain feedback from specialists and junior physicians. The final version of the questionnaire consisted of 13 questions; eight discrete choice questions, three 5-Point Likert scale questions, and two 6-Point Likert scale questions. A knowledge score (KS) of 0 to 100% was calculated based on participants’ agreement or disagreement with seven statements (available in the Appendix A). The mean of correct statements was calculated for each participant as well as the maximum, minimum, standard deviation, and mean of correct answers for all participants.

The items in each of the 13 questions were randomized to avoid order bias and a completeness check was performed automatically by SurveyMonkey at the end of the survey, once respondents had clicked the “Submit” button. All items in each question were mandatory and no non-response options such as “not applicable” or “prefer not to say” were available. Respondents were able to review and edit their answers, provided they did not click on the “Submit” button to exit the survey. The answers were automatically captured by SurveyMonkey and then extracted onto an Microsoft Excel database (version 16, Microsoft Corporation, Redmond, USA). The data was collected between 25 January 2022 and 25 May 2022 and IP addresses and cookies were not recorded as the survey was guaranteed to be anonymous.

As our survey did not collect any personal or healthcare data, and was strictly anonymous, the local Ethics Committee (Commission cantonale d’éthique de la recherche sur l’être humain) ruled that it did not require further approval (Req. n°2021–00773) [38]. Participants were informed of the terms of the study (i.e., anonymity, purpose, data processing, and identity of the investigator) through an informative introductory page at the beginning of the survey and by their respective head physicians. Rheumatologists and gastroenterologists were also informed by their professional association via e-mail. Furthermore, this study was written in accordance with the Checklist for Reporting Results of Internet E-Surveys (CHERRIES) statement [39].

Descriptive statistics were reported using numbers, means, and standard deviations. Analysis of the variance of the dependent variables was performed using Bartlett’s and Levene’s tests. Parametric and nonparametric methods were used to compare the participants’ age, KS, and years of cumulative experience (CE). Age was assessed by specialty and place of practice. Years of CE were compared by specialty and place of practice, and KS was evaluated by specialty, years of CE and place of practice. Ordinal data issued from the 5 and 6-Point Likert scale questions were evaluated by specialty, years of CE, and place of practice using nonparametric methods. The association between the participants’ answers to the 5 and 6-Point Likert scale questions and the KS was assessed using ordinal regression analysis. Statistical significance was considered at a *p* < 0.05 threshold. All analyses were performed using R (Version 4.2.0, R Foundation for Statistical Computing, Vienna, Austria) and are available in Appendix A [40].

## 3. Results

### 3.1. Respondent’s Characteristics

A total of 37 physicians exclusively from the cantons of Geneva and Vaud participated in the survey, representing 10% of the specialists in these cantons as of May 2022, according to the Swiss Medical Register: four (11%) in gastroenterology, 11 (30%) in immunoallergology and 22 (59%) in rheumatology [41]. Thirty (81%) participants were primarily in hospital practice, while the remainder were in office practice. The mean (sd) age was 44 (12) years, with rheumatologists being significantly older than gastroenterologists and immunoallergologists (*p* < 0.01). Office-based physicians were also significantly older than hospital-based physicians (*p* < 0.01). Eighteen participants (49%) reported having less than 10 years of cumulated experience (CE), eight had between 10 and 20 years of CE (22%) and 11 had more than 20 years of CE (30%).

The most frequently treated diseases were psoriatic arthritis (*n* = 20, 17%), rheumatoid arthritis (*n* = 20, 17%), and systemic inflammatory diseases (*n* = 18, 16%). Four physicians (11%) had not yet prescribed TNF-⍺ antagonists biosimilars at the time of the survey. The reference TNF-⍺ antagonists whose biosimilars were the most prescribed were: infliximab (22, 71%), etanercept (16, 52%), and adalimumab (15, 48%). The prescription frequency of TNF-⍺ antagonist biosimilars is shown in Figure 1.

### 3.2. Experience and Knowledge Regarding Biosimilars

A majority of respondents (*n* = 25, 67%) agreed or strongly agreed (strongly/agreed) that their level of knowledge regarding biosimilars was “good”. Similarly, participants strongly/agreed (*n* = 26, 70%) that they felt “well informed” about biosimilars, having enough information about safety of use (n strongly/agree = 27, 82%) and therapeutic efficacy (n strongly/agree = 26, 79%) to be comfortable with prescribing them. With respect to the level of knowledge and information, office-based physicians agreed significantly more than hospital-based practitioners (*p* < 0.05).

The most common sources of information used by participants were “the Swiss Compendium of Medicines website” and “self-study and scientific literature” (Figure 2) [42]. The least used source of information were “the media” and physicians’ “fellow pharmacists”. Rheumatologists received more visits from industry representatives than gastroenterologists and immunoallergologists (*p* = 0.03).

Twenty-eight (85%) participants strongly/agreed that they present information about biosimilars in a positive manner when discussing them with their patients. However, only a minority explicitly use the term “biosimilar” during consultations with their patients (n disagreed or strongly disagreed = 13 (39%), n neither disagreed nor agreed = 9 (27%), n strongly/agreed = 11 (33%)), with rheumatologists using the term “biosimilar” significantly more than gastroenterologists (*p* < 0.05).

Twenty-five participants (75%) strongly/agreed that their patients who have not yet started treatment with the reference biologic (i.e., originator) were willing to start treatment with the biosimilar. Twenty-three participants (69%) strongly/agreed that they had enough time to offer a biosimilar to their biologic-naive patients (BNPs) and explain the reasons why. Furthermore, a majority (*n* = 23, 69%) of physicians strongly/agreed with routine prescribing of biosimilars instead of original biologics for BNPs. Additionally, 28 (84%) physicians strongly/agreed that patients who have never used biologics should start treatment with a biosimilar, if one exists and is available. On the contrary, only 5 (15%) participants agreed that their patients in remission were willing to substitute their originator with the biosimilar, and the majority (*n* = 17, 51%) disagreed or strongly disagreed (strongly/disagreed) with having enough time to propose a biosimilar substitution and explain the purpose.

#### Knowledge Score (KS)

The average (sd, min, max) KS score of all participants was 74% (16, 29%, 100%). There were no significant differences in mean KS between gastroenterologists, immunoallergologists, and rheumatologists, but participants with more than 20 years of CE had a significantly lower KS than respondents with between 10 and 20 years of CE (64% vs. 84%, *p* = 0.02). The lowest rate of correct answers (*n* = 17, 46%) was for the statement “A biosimilar has the same immunogenicity as its originator”. The rate of correct answers for the three statements regarding the definition of biosimilars was 78%, 68%, and 68% (Figure 3).

In an ordinal regression model, there was a significant negative association between participants with higher KS and the statement “I prescribe a biosimilar in a patient based primarily on my clinical experience” (*p* = 0.02). In a second ordinal regression model, there was a significant negative association between KS and the statement “I explicitly use the term “biosimilar” when talking to my patients who are going to start a biosimilar or to whom I want to suggest a biosimilar substitution” (*p* = 0.02). On the contrary, there was a significant positive association between KS and participants agreeing with the statement “The lack of incentive systems is a barrier to the prescription of biosimilars in Switzerland” (*p* = 0.02)

### 3.3. Confidence in TNF-⍺ Antagonist Biosimilars

Only a few physicians disagreed (*n* = 4, 11%) or strongly disagreed (*n* = 1, 3%) with being comfortable talking about the benefits of biosimilars to their patients or colleagues. Rheumatologists felt more comfortable than immunoallergologists and gastroenterologists in both cases (*p* < 0.01), while immunoallergologists were more comfortable than gastroenterologists when discussing biosimilars with their patients (*p* < 0.01). A slight majority of participants (*n* = 11, 33%) strongly/agreed that they prescribed biosimilars based primarily on their own clinical experience. Similarly, a larger majority (*n* = 19, 57%) strongly/disagreed with the statement “When a biosimilar comes on the market, I prefer to wait for the results of substitution in my colleagues’ patients before proposing the substitution to my patients”. The majority of physicians (*n* = 30, 90%) readily prescribed biosimilars to BNPs using literature findings, with rheumatologists more likely than immunoallergologists and gastroenterologists (*p* < 0.01). Thirty (90%) participants expressed confidence in the therapeutic management of BNPs for whom they prescribed biosimilars. Furthermore, most participants (*n* = 26, 79%) strongly/agreed with initiating treatment with a biosimilar rather than substituting the originator in patients already in remission.

Sixteen (48%) physicians strongly/disagreed to offer a biosimilar substitution to their patients already on an originator and in remission. If participants relied solely on findings from the literature, the majority (*n* =19, 57%) strongly/disagreed to switch patients who had difficulty achieving remission. Yet, seventeen (51%) participants strongly/agreed with the statement “Prescribing a biosimilar gives me confidence in the therapeutic management of my patients who have not yet started their treatment with the originator”.

### 3.4. Opinion Regarding Biosimilars of TNF-⍺ Antagonists

Biosimilar prescribing was well promoted in the participant's institutions/office (n strongly/agreed = 22, 63%). However, when asked if biosimilar prescription was promoted by the Swiss Federal Office of Public Health, participants’ answers were mixed (n strongly/disagreed = 11 (33%), n neither disagreed nor agreed = 11 (33%), n strongly/agreed = 11 (33%)). On another note, a minority (*n* = 4, 12%) acknowledged that they had refrained from prescribing biosimilars because health insurance companies routinely blocked their reimbursement for off-label use. Thirty-one (94%) physicians strongly/agreed that biosimilars represent an opportunity to reduce healthcare costs. Nevertheless, opinions were more mitigated regarding the statement “The lack of incentive systems is a barrier to the prescription of biosimilars in Switzerland” (n strongly/disagreed = 6 (18%), n neither disagreed nor agreed = 15 (39%), n strongly/agreed = 14 (42%)). Answers were mixed when participants were asked if the success of a substitution of an originator by its biosimilar depended mainly on the physician-patient relationship (*n* strongly/disagreed = 6 (18%), *n* neither disagreed nor agreed = 10 (30%), *n* strongly/agreed = 17 (51%)).

## 4. Discussion

To our knowledge, this is the first study assessing the expertise, experience, and opinion regarding TNF-⍺ antagonist biosimilars of postgraduate physicians and specialists in rheumatology, gastroenterology, and immunoallergology in Western Switzerland. This study follows previous work undertaken in Western Switzerland, which reported a substantial discontinuation rate among patients treated with an infliximab biosimilar (i.e., CT-P13) [34]. Given the economic stakes associated with biosimilars, an evaluation of potential barriers to the use of biosimilars in Western Switzerland was required.

### 4.1. A Slow but Positive Trend in Knowledge and Confidence in Biosimilars

In general, the physicians in our study felt well-informed about biosimilars and agreed that their level of knowledge was “good”. They felt comfortable discussing biosimilars with their colleagues and patients, and reported positively presenting information about biosimilars, even if they did not explicitly use the term “biosimilar” during consultations with their patients. Their good level of knowledge was objectively confirmed with the calculation of a knowledge score (KS). However, questions regarding the technical definitions related to the structure of biosimilars (i.e., amino acid chain and glycosylation) or to their similarity to their respective original biologics (originators) were not answered correctly by a majority of our participants, reflecting the results reported by a 2021 systematic review in which physicians tended to have higher self-reported knowledge compared with their actual knowledge [21]. Furthermore, only 46% of our participants stated that biosimilars had the same immunogenicity as their originators, which is also consistent with the literature, where varying levels of surveyed physicians believe that a biosimilar and its originator do not have the same immunogenicity [19,36,43,44]. In addition, KS was negatively associated with using clinical experience alone to prescribe biosimilars, meaning that participants with higher KS relied on other sources of information to guide their decision (e.g., literature or fellow physicians).

Physicians practicing for more than 20 years had a significantly lower KS than those who had between 10 and 20 years of cumulated experience. Of course, this result alone does not suggest that younger generations of physicians are better trained and informed about biosimilars, but a trend may be emerging as new biosimilars are being approved. A comparison of the data from two studies from the Alliance for Safe Biologic Medicines (ASBM) that took place in 2013 and 2019 revealed a 76% to 90% increase in the level of knowledge about biosimilars in European prescribers [27]. This observation is consistent with the hypothesis that there is a positive trend in the positive evolution of knowledge about biosimilars over the past decade. Interestingly, the 2019 ASBM study also reported that 31% of Swiss physicians considered themselves “very familiar” with biosimilars, 51% had only a “basic understanding”, 14% “could not define them”, and 5% had “never heard of them”. It is worth noting that a higher proportion of physicians in our study agreed that they had a “good” level of knowledge regarding biosimilars, compared to the 2019 ASBM’s data.

Three older surveys conducted between 2014 and 2015 reported lower awareness of biosimilars and limited confidence in prescribing them, certainly due to the limited availability of biosimilars at that time [17,19,20]. The first study (2014) was conducted in Europe with 307 inflammatory bowel disease (IBD) specialists and reported that 61% felt “little or no confidence” in prescribing them [19]. The second one (2015) was led by the ASBM in South America with 399 prescribers from four countries and 10 specialties [17]. Again, only 53% of the participants considered themselves “familiar” with biosimilars. The third one (2015), with 220 Japanese rheumatologists and oncologists, reported that 35% of them had “never heard” of the term “biosimilar” [20]. This relative lack of awareness during these years may be explained by the fact that, until 2016, only four biologics (i.e., epoetin alfa, filgrastim, insulin glargine, and somatropin) had biosimilars approved by the European Medicines Agency (EMA), the world’s primary regulatory body for biosimilar drug approval [45,46,47,48,49]. However, despite multiple biosimilars being approved in the following years (TNF-⍺ antagonists: infliximab (2016), etanercept (2017), and adalimumab (2017); antineoplastic agents: rituximab (2018), trastuzumab (2018), and bevacizumab (2019)), physicians’ global expertise in biosimilars remained low, while older concerns flourished, suggesting that the positive trend mentioned above may take more years to realize its potential [50,51,52,53,54,55].

Hence, following these European market authorizations, two systematic reviews (from 2019 and 2021) summarized the state of physicians’ knowledge and acceptance of biosimilars in multiple specialties. The first included three U.S.-based studies and 17 that originated in Europe (from 2014 to 2018) [22]. The second included 23 studies from 2014 to 2019, also primarily European [21]. Both reviews reported varying levels of familiarity with biosimilars (49% to 67%) and general concerns about the safety, efficacy, extrapolation, interchangeability, and immunogenicity of biosimilars. The second review also reported that physicians’ objectively measured knowledge, was generally more limited compared to their self-reported knowledge, as mentioned before. In addition, the most recent studies from 2021 reported that efficacy and confidence in biosimilars were still the primary barriers to their prescribing, in multiple settings and specialties [24,29,31].

### 4.2. Physician’s Prescription Behavior

Regarding their prescription behavior, physicians in our study preferred to prescribe a biosimilar to BNPs rather than substituting the originator for an already stabilized patient, which is also reported by multiple studies with participants from various specialties and settings in the literature [24,31,56,57]. Interestingly, participants in our study were less likely to switch stabilized patients than the 95 Swiss physicians in the 2019 ASBM study [27]. This may be because the physicians in our study reported not having enough time to manage biosimilar substitutions or because their patients, who were already taking the originator, were reluctant to consider a non-medical switch (i.e., a change of medication in a stable patient to another (non-generic) medication, for reasons other than side effects, poor adherence, or lack effectiveness) [58]. Another explanation comes from several observational studies in the literature which reported that patients who were switched from an originator to its biosimilar experienced an increase in their disease symptoms, a loss of treatment efficacy or even adverse effects [59]. In any case, this raises the question of the extent to which prescribers are aware of their contribution to healthcare costs, as biosimilars have to be at least 25% less expensive than their originators in Switzerland. In our study, the positive association between KS and the statement *“The lack of incentive systems is a barrier to the prescription of biosimilars in Switzerland*” suggests that physicians with greater knowledge regarding biosimilars are more sensitive to the economic implications of biosimilars and their prescription. This is not necessarily a given in Switzerland, as originator drugs and their biosimilars benefit from the same reimbursement rates and patients contribute only 10% of their outpatient treatment in direct payments.

In the rest of the literature, specialists’ prescription behavior reported by studies from 2019 onward diverged by location. Indeed, European-based studies tended to report a greater proportion of physicians who are comfortable with biosimilar substitution or are likely to switch patients who are doing well on the originator, compared with countries outside EMA jurisdiction or under FDA regulations.

### 4.3. Opinion Regarding the Prescription of TNF-⍺ Antagonists Biosimilars in Switzerland

Promoting the prescription of biosimilars at the national level, for example through incentives or national tenders, is complex because Switzerland is a federal state (confederation). Hence, each member state (cantons) retains its independence but is subject to the central authority, essentially consisting of a coordinating body whose decisions must be taken unanimously by the member states. In general, although the central authority coordinates health policy strategies (e.g., strategy “Health2030”), the cantons have the freedom to implement them as they see fit [60]. Hence, even though the prescription of TNF-⍺ antagonist biosimilars is known to be well-promoted in Western Switzerland, it is not necessarily the case in the central and eastern part of Switzerland, as reported by the Swiss Biosimilars Barometer (SBB), an annual report published by a community of interest to promote biosimilars in Switzerland [61,62]. This year’s report noted that regional disparities in Switzerland regarding the prescribing of biosimilars continued to widen and that drug costs for mandatory health insurance could have been reduced by about 100 million Swiss francs if biosimilars had been used consistently [61,63]. Although this amount of savings is based only on direct purchase prices and does not take into account discounts negotiated between hospitals and pharmaceutical industries or the full direct costs required to introduce a biosimilar in a hospital, it does highlight the political power of cantons to operate as they see fit and to prioritize the use of more expensive originators in their hospitals [34]. Indeed, there were no official guidelines for the implementation, management, or use of biosimilars in Switzerland at the time of this study.

The main limitation of the present study is its relatively low response rate and thus its ability to infer to a larger population of rheumatologists, gastroenterologists, and immunoallergologists in the cantons of Vaud and Geneva. However, the majority of responses were rarely correlated with specialties (i.e., rheumatology, gastroenterology, or immunology), years of cumulative experience (i.e., <10 years, between 10 and 20 years, or >20 years), or place of practice (i.e., office or hospital), meaning that the overall results reported by our study might be generalized to all practitioners prescribing TNF-⍺ antagonist biosimilars in Western Switzerland.

Future studies should include the rest of the regions of Switzerland, or at least the cantons with a university-level hospital (i.e., Basel, Bern, and Zurich) to better characterize the regional differences identified in the SBB report. Creating a questionnaire that would be fully understood by all prescribers of biosimilars in Switzerland is a complex task; not only would there be specificities with respect to the prescribers’ specialties or place of practice, but also back-translation methodologies (two-way translations by two independent translators) would be required to ensure that the questionnaire is understood in French, German and Italian (the three official languages in Switzerland.). Nevertheless, such results would help to understand and address regional disparities regarding the use of TNF-⍺ antagonist biosimilars in Switzerland and lay the groundwork for concrete actions to optimize our healthcare resources.

## 5. Conclusions

The present study reported current expertise, opinions, and attitudes toward TNF-⍺ antagonist biosimilars among postgraduate physicians and specialists in Western Switzerland. Participants’ knowledge and confidence in these biosimilars were high and consistent with the current trend reported in the literature. Respondents seemed particularly comfortable initiating biosimilar therapy in biologic-naive patients, whereas a significant proportion expressed concerns about switching patients already receiving an originator. Further studies would be needed to assess the need for specific guidelines for biosimilar switching or whether national incentives should be established to facilitate the communication around these biosimilars and their prescription.

## Figures and Tables

**Figure 1 healthcare-10-02152-f001:**
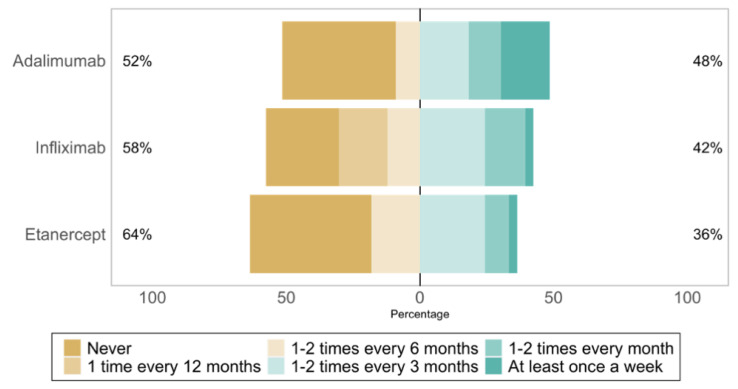
Prescribing frequency of reference TNF-⍺ antagonists biosimilars, from most to least prescribed (*n* = 33).

**Figure 2 healthcare-10-02152-f002:**
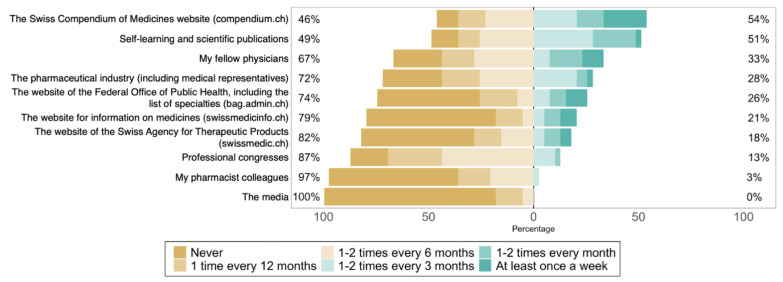
Participants’ answers to each item in the question “In the past 12 months, how often have you used the following sources of information regarding biosimilars?” (*n* = 37).

**Figure 3 healthcare-10-02152-f003:**
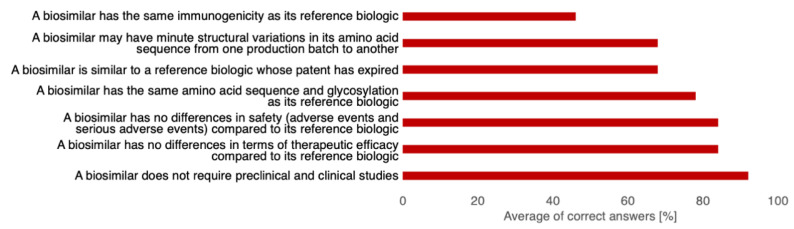
Averages of correct answers for each statement used to calculate the knowledge score. (*n* = 37).

## Data Availability

Dataset and questionnaire available upon request to the corresponding author.

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
