# Peer review of "Current Expertise, Opinions, and Attitude toward TNF-⍺ Antagonist Biosimilars among Physicians: A Self-Administered Online Survey in Western Switzerland"

_healthcare, 2022, doi:10.3390/healthcare10112152_

Round 1
Reviewer 1 Report
I read with interest this manuscript offering a large overview on such an important topic.
I found the paper well written and easily readable, although some long lists of percentages could a little bit affect the readers' involvement.
Statical methods seems to be soundy, and the effort to have a good compliance among the cathegories to whom the questionnaire was addressed to was extensive.
Graphics are easy to read and well constructed.
I have just two questions:
1) Authors stated that their questtionnair was addressed to physicians and specialists of different medicine branches "managing patients eligible for TNF-⍺ antagonists and their biosimilars." How the Authors can know which of them actually manage those patients? If there is a source of data for this selection, it shuold be mentioned. If not, and the questionnaire was address to all these physicians and specialists because it was likely that they manage patients eligible for TNF-⍺ antagonists and their biosimilars, I suggest to rephrase that sentence.
2) This study has at least one limitation in the response rate, that for some specialists was very low (i.e. for instance, gastroenterologists). This issue could have affected results? I think the Authors should discuss study limitations, including the one I remarked, in their discussion.
Author Response
We would first like to thank the reviewer for his careful proofreading and suggestions for improvement.
We kindly ask the reviewer and the editor to find our answers, point by point, below:
1) Authors stated that their questionnaire was addressed to physicians and specialists of different medicine branches "managing patients eligible for TNF-⍺ antagonists and their biosimilars." How can the Authors know which of them actually manages those patients? If there is a source of data for this selection, it should be mentioned. If not, and the questionnaire was addressed to all these physicians and specialists because it was likely that they manage patients eligible for TNF-⍺ antagonists and their biosimilars, I suggest to rephrase that sentence.
Response 1): In a cohort of patients with gastroenterological disorders, rheumatological disorders, or immuno-allergic disorders, all patients are likely to be eligible for TNF-alpha antagonist therapy. As the reviewer pointed out the questionnaire was addressed to all these physicians and specialists because it was likely that they manage patients eligible for TNF-⍺ antagonists and their biosimilars. We understand that this long first sentence in Chapter 2, "Materials and Methods," is not clear. You will find the following rewording in the manuscript:
Old phrase: "The questionnaire was directed to post-graduate physicians and specialists in rheumatology, gastroenterology, and immunoallergology exercising in Western Switzerland and managing patients eligible for TNF- ⍺ antagonists and their biosimilars."
New phrase: "The questionnaire was sent to post-graduate physicians and specialists in rheumatology, gastroenterology, and immunoallergology practicing in Western Switzerland, likely to prescribe TNF-⍺ antagonists and their biosimilars."
2) This study has at least one limitation in the response rate, that for some specialists was very low (i.e. for instance, gastroenterologists). This issue could have affected results? I think the Authors should discuss study limitations, including the one I remarked, in their discussion.
Response 2): We agree with the reviewer. In response to his pertinent remark, we have added a paragraph, previously deleted, in chapter 4.3 of the discussion: "The major limitation of the present study is its relatively low response rate and thus its ability to infer to a larger population of rheumatologists, gastroenterologists, and immunoallergologists in the cantons of Vaud and Geneva. However, the majority of responses were rarely correlated with specialties (i.e., rheumatology, gastroenterology, or immunology), years of cumulative experience (i.e., <10 years, between 10 and 20 years, or >20 years), or place of practice (i.e., office or hospital), meaning that the overall results reported by our study might be generalized to all practitioners prescribing TNF-⍺ antagonist biosimilars in Western Switzerland."
Reviewer 2 Report
The study aimed to evaluate the knowledge, opinion, and attitude toward TNF-⍺ antagonist biosimilars among post-graduate physicians and specialists from Western Switzerland.
The manuscript is well written, but I recommend that the authors clarify in the discussion section (4.2. Physician’s prescription behavior) the risks associated with switching patients from reference medicines (originators) to biosimilars.
Spaces are missing in many places before bibliographic references.
Author Response
We would like to thank the reviewer for his suggestion for improvement. We have therefore added any missing spaces before the opening brackets of the references.
We also kindly ask the reviewer and the editor to find our response below:
1) The manuscript is well written, but I recommend that the authors clarify in the discussion section (4.2. Physician’s prescription behavior) the risks associated with switching patients from reference medicines (originators) to biosimilars.
Response 1): As suggested, we have added a brief explanation of the risks associated with switching patients from brand-name drugs to biosimilars and a relevant citation: "Another explanation comes from several observational studies in the literature which reported that patients who were switched from an originator to its biosimilar experienced an increase in their disease symptoms, a loss of treatment efficacy or even adverse effects [59]."
59 Cohen, H. P. et al. Switching Reference Medicines to Biosimilars: A Systematic Literature Review of Clinical Outcomes. Drugs 78, 463-478 (2018). https://doi.org:10.1007/s40265-018-0881-y
Reviewer 3 Report
This paper highlights "Current expertise, opinions, and attitude toward TNF-⍺ antagonist biosimilars among physicians, which is based on a self-administered online survey in Western Switzerland. In an era where generics and Biosimilars are of high need to address the issue of high prices of reference biologic (i.e. originator) for the treatment of certain diseases such as cancers. The authors to their best of their ability stressed out the the knowledge and prescription opinion of three FDA-approved TNF-⍺ antagonists (infliximab, etanercept and adalimumab). However; despite all the reasons given in this paper, there is a economic and Insurance coverage aspect that has not be extensively discussed. For instance, will the difference of prices and Insurance coverage percentage be a reason that could boost the perception and prescription of these biosimilars among physicians? I do believe that discussing this financial aspect of could enhance this paper scientific soundness.
Author Response
We would like to thank the reviewer for his pertinent question regarding the reimbursement aspects surrounding TNF-⍺ biosimilars and the possible economic incentives that may exist. We kindly ask the editors and the reviewer to find our answer below.
1) For instance, will the difference of prices and Insurance coverage percentage be a reason that could boost the perception and prescription of these biosimilars among physicians? I do believe that discussing this financial aspect of could enhance this paper scientific soundness.
Response 1): Without being able to answer the reviewer's question perfectly, the influence of costs relative to originators and biosimilars was addressed in chapter 4.2 "physician's prescription behavior" by these few lines:"(…) In any case, this raises the question of the extent to which prescribers are aware of their contribution to health care costs. In our study, the positive association between KS and the statement “The lack of incentive systems is a barrier to the prescription of biosimilars in Switzerland” suggests that physicians with greater knowledge regarding biosimilars are more sensitive to the economic implications of biosimilars and their prescription."
Since there are currently no tangible studies in Switzerland on the link between originator and biosimilar prices and doctors' prescribing practices, or political incentives to lower originator reimbursement rates, we have modified the text as follows to best incorporate the reviewer's suggestions. Additional explanations in the original text are underlined below.: "(…) In any case, this raises the question of the extent to which prescribers are aware of their contribution to health care costs, as biosimilars have to be at least 25% less expensive than their originators in Switzerland. In our study, the positive association between KS and the statement “The lack of incentive systems is a barrier to the prescription of biosimilars in Switzerland” suggests that physicians with greater knowledge regarding biosimilars are more sensitive to the economic implications of biosimilars and their prescription. This is not necessarily a given in Switzerland, as originator drugs and their biosimilars benefit from the same reimbursement rates and patients contribute only 10% of their outpatient treatment in direct payments."
Reviewer 4 Report
Manuscript entitled "Current expertise, opinions, and attitude toward TNF-⍺ antagonist biosimilars among physicians: A self-administered online survey in Western Switzerland" authored by Marko Krstic et al., conducted a survey to understand knowledge, attitude and prescribing pattern of three TNF-⍺ antagonist biosimilars, infliximab, etanercept and adalimumab. These biologics are used to treat inflammatory autoimmune diseases like rheumatoid arthritis, psoriatic arthritis, inflammatory bowel disease and immuno-allergy diseases. The method used for data collection in this survey is based on the 13 questions to the postgraduate physician in the western Switzerland as supported by the supplementary data. Current manuscript is a well written and presented study. The results of this study may be used to formulate guidelines for the use of biosimilars at national levels. The present study is a flaw less and therefore qualify for publication in Healthcare.
Author Response
We would like to thank the reviewer for his careful proofreading and very positive comment on our study and article.
We have carefully re-read our article to clarify some of the sentence structure and ambiguities.